# Effects of maternal healthcare service utilization on modern postpartum family planning access in Bangladesh: insights from a National representative survey

**Md. Nuruzzaman Khan**[1,2*�], **Md. Mostaured Ali Khan**[3�], **Md Arif Billah**[4�], **Shimlin Jahan Khanam**[1], **Md. Moinuddin Haider**[4], **Bidhan Krishna Sarker**[3‡], **Melissa L. Harris**[2‡]

**1** Department of Population Science, Jatiya Kabi Kazi Nazrul Islam University, Mymensingh, Bangladesh, **2** Nossal Institute for Global Health, Melbourne School of Population and Global Health, The University of Melbourne, Australia, **3** Maternal and Child Health Division, International Centre for Diarrhoeal Diease Research, Dhaka, Bangladesh, **4** Health System and Population Studies Division, International Centre for Diarrhoeal Diease Research, Dhaka, Bangladesh

he Joint 1st author as they contributed equally to this paper.
‡ Joint Senior authorship.
* sumonrupop@gmail.com

## Abstract

### Background

Access to modern family planning is critical for improving maternal and child health outcomes, yet it remains severely lacking in low- and middle-income countries, including Bangladesh. Maternal healthcare utilization during and after pregnancy is vital for promoting postpartum family planning. This study examined the effects of maternal healthcare service utilization on postpartum family planning uptake in Bangladesh.

### Methods

Reproductive calendar data from 4,081 women with recent live births were extracted from the cross-sectional 2017/18 Bangladesh Demographic and Health Survey and analyzed. The outcome variable was uptake of modern postpartum family planning methods and the exposure variables were different types of maternal healthcare services. Kaplan-Meier methods were used to calculate cumulative probabilities of modern postpartum family planning method uptake within 12 months post-delivery, and modified Poisson regression models were used to estimate the effects of utilizing maternal healthcare services on modern postpartum family planning method uptake.

### Results

Modern family planning methods in the 12 month postpartum period were used by 72% of women, with over 60% starting after day 40. Less than 4% used long-acting family planning methods, while almost 40% relied on the oral contraceptive pill. Utilizing maternal

**Data availability statement:** The datasets used and analyzed in this study are available from the Measure DHS website: https://dhsprogram.com/data/available-datasets.cfm.

**Funding:** The author(s) received no specific funding for this work.

**Competing interests:** The authors have declared that no competing interests exist.

**Abbreviations:** LMICs, Low- and Middle-Income Countries; DHS, Demographic Health Survey; BDHS, Bangladesh Demographic Health Survey; PPFP, Postpartum family planning; ANC, Antenatal care; PNC, Postnatal care; CI, Confidence Interval; SDG, Sustainable Development Goal; NIPORT, National Institute of Population Research and Training; PSU, Primary Sampling Unit.

healthcare services was associated with up to a 7% higher uptake of modern postpartum family planning methods compared to non-users.

## Conclusion

Three-quarters of Bangladeshi women use modern family planning within the 12 months postpartum, but often rely on less effective methods. Additionally, 25% of these women resort to traditional or no use of contraceptive methods, increasing the risks of unintended pregnancy, short birth intervals, and adverse maternal and infant health outcomes. Maternal healthcare services, including private facilities, should prioritize modern postpartum family planning provision, along with education and counseling on the benefits of long-acting contraception.

## Background

Sustainable Development Goal (SDG) 3 aims to significantly reduce preventable maternal and child mortality which is widespread in low- and middle-income countries (LMICs) [1]. Consistent and correct use of family planning (FP) methods can reduce the occurrence of maternal and child mortality in LMICs [2] through reductions in short-interval births and unintended pregnancies. However, these issues remain persistent public health challenges in LMICs accounting for 25% and 49% of total pregnancies, respectively [3]. Notably, in the extended postpartum period (i.e., 0–12 months following delivery), the non-use of FP methods has been associated with all short interval births and the majority of the unintended pregnancies in LMICs [3]. Therefore, increasing the use of modern postpartum family planning (PPFP) methods, such as progestin-only pills, injectables, implants, and intrauterine devices (IUDs), condoms, is crucial to preventing these adverse outcomes.

Despite the urgent need for modern PPFP, only 41% of women in LMICs use such methods during the postpartum period [4]. This burden is additional to the ongoing lower coverage of modern contraception use (58%) seen in LMICs [4]. Importantly, women have been found to have a higher unmet need for FP methods during the postpartum period (49%) compared to the preconception period (24%). This unmet need is even higher in South Asian countries (59%) [4]. This is despite 95% of women intending to either not have another child or delay their next pregnancy [5]. These figures indicate potential barriers during the postpartum period that hinder women's access to modern PPFP methods. Consequently, the World Health Organization (WHO) has advocated for the use of modern FP methods in the postpartum period, and this issue has been prioritized as a key indicator of national FP programmes in LMICs including Bangladesh [2].

Bangladesh has one of the highest rates of maternal (173 per 100,000 livebirths) and under-five mortality (45 per 1,000 livebirths) among LMICs [6,7] which is exacerbated by high rates of unintended pregnancy (47%) [8] and short-interval births (26%) [9]. Higher occurrences of unintended pregnancy and short birth interval have been found to occur mostly due to a very high rate of non-use (46%) and unmet need for FP methods (12%) [10,11]. Unmet need for FP methods is considered to be higher in the postpartum period; however, an accurate estimate is lacking.

Unfortunately, these situations have remained unchanged in Bangladesh since the early 2000s, despite the government's efforts to improve maternal and infant outcomes in line with the Millennium Development Goals [7]. These included strength in FP services at the field level and increased education and awareness regarding FP. However, the lack of progress in Bangladesh instigated the development of the National Action Plan for PPFP [10]. As part

of this plan, coordination between healthcare and FP service providers was established to increase the uptake of FP methods during the postpartum period through counselling and provision of modern FP methods during maternal healthcare service attendance [10]. These services provide numerous opporunties to have discussions about modern PPFP methods during antental care, delivery and postpartum care. Such programs have been provided in addition to previous home-based contraceptive approaches. A similar approach (i.e counselling abour the importance of PPFP during maternal healthcare services uptake) is also recommended by the WHO to improve PPFP uptake at a global level [12] and has been found to be effective in LMICs [13–15]. However, regardless of this focus for PPFP, related research are scare in Bangladesh and LMICs mostly because of lack of data. Current research primarily focuses on low use of modern contraception (54%) and overall unmet need for contraception (12%) and their associated socio-demographic factors [16–19], as is the case in other LMICs [20,21]. We therefore aimed to determine the effects of maternal healthcare services use, including antenatal care (ANC), delivery with skilled birth attendants (SBA), delivery care in healthcare facilities (DHC), caesarean section delivery and postnatal care (PNC), on modern PPFP methods uptake in Bangladesh.

## Methods

### Data source

This study analysed national representative cross-sectional survey data from the 2017–18 Bangladesh Demographic Health Survey (BDHS), conducted as part of the Demographic and Health Survey (DHS) program. Data collection took place between September and December 2017. The National Institute of Population Research and Training (NIPORT) conducted this survey in Bangladesh as a local partner and as a representative of the Ministry of Health and Family Welfare of Bangladesh (MoHFW). A two-stage stratified random sampling procedure was used. The first stage involved randomly selecting 675 primary sampling units (PSUs). This included the 293,579 PSUs listed by the Bangladesh Bureau of Statistics (BBS) as part of the 2011 National Population Census. Of these primary selected PSUs, data collection was undertaken in 672 PSUs with the remainder excluded due to flood. In the second stage of sampling, 30 households were randomly selected from each PSU, generating a list of 20,160 households covering 20,376 eligible respondents. Interviews were conducted in 19,457 households (96% of coverage). There were 20,376 eligible women in the chosen households, and data were gathered from 20,127 women with a response rate of >99%. The inclusion criteria for completion of this survey were: (i) ever-married women aged 15–49 years and (ii) staying in the selected households on the night preceding the survey. Eligible women were asked about their socio-demographic and reproductive health-related characteristics, such as the use of ANC, DHC, and PNC for their most recent pregnancy that occurred within three years of the survey date. BDHS also collected information on FP method use, pregnancy, live births, and termination history for up to five years preceding the survey. Further details regarding the sampling strategy have been published elsewhere [22].

### Study participants

This study focused on ever married women aged 15–49 years who met three specific criteria: i) had given birth within three years prior to the survey, ii) had data available on their use or non-use of family planning methods during the postpartum period and maternal healthcare services, and (iii) had completed the postpartum period (up to 12 months from delivery as per the WHO recommendation). In the 2017–18 BDHS, a total of 7,562 women were interviewed who had given birth within five years. Of these 5,012 women had given birth within the previous three years of the survey and had data on family planning method use, as well as maternal

healthcare service utilization. From these, data from 931 women were further excluded because they were either pregnant (n = 179) or were currently in the postpartum period (n = 752). A total of 4,081 women met the participant selection criteria for inclusion in this analysis (Fig 1).

## Outcome variable

The primary outcome variable for this study was modern PPFP method use during the postpartum period (i.e., 0–12 months post-delivery, as recommended by the WHO) [1]. To generate this variable, we used women's reproductive calendar data, taking into account the month they started using FP methods following a live birth and the types of FP methods used. In this particular form of data collection technique, women were asked to document their monthly history of contraception use or non-use by asking respondents monthly history of contraception during the last few years (asked about the form of contraception use and duration of use and consolidated information was reported) and provide reasons for non-use or discontinuation.

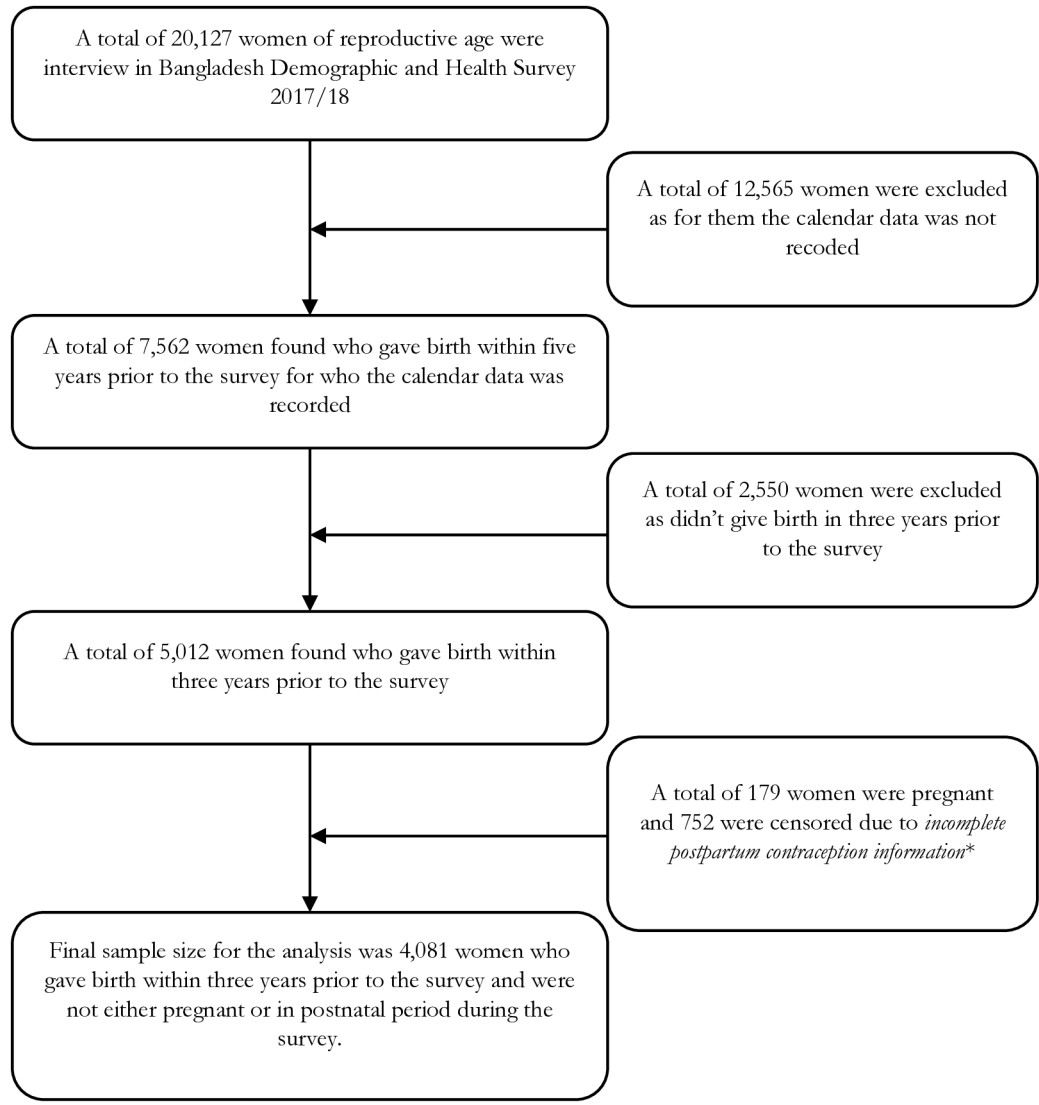

**Fig 1. Selection of the study participations following the STROBE guideline.**

A list of contraceptive methods available in Bangladesh at the time of the survey, as outlined in Table 2, were provided to assist reporting. Respondents were also given the option to indicate if the contraceptive methods they used were not listed. We reclassified these responses according to WHO's categorization guidelines into the utilization of modern FP methods (which included the pill, intra-uterine devices (IUDs), injectables, condoms, female sterilization, male sterilization, and implant, coded as 1) and other methods (coded as 0). We also extracted data on the duration (in months) of initiating modern FP use. This was measured from the date of the most recent childbirth within the first 12 months postpartum period and censored at the time of FP discontinuation or at 12 months whichever occurred first. Duration of PPFP was defined as the month in which modern contraception was first initiated after the live birth.

## Exposure variables

Utilization of maternal healthcare services was our primary exposure of interest. Based on the WHO guidelines these included ANC (0 = no, 1 = yes), SBA (0 = no, 1 = yes), DHC (0 = no, 1 = yes), caesarean delivery (0 = no, 1 = yes), and PNC (0 = no, 1 = yes). The WHO's 2014 guideline stated that during the course of a pregnancy, each woman should: i) receive at least four skilled ANC visits (while this was revised to eight ANC visits in 2016, Bangladesh's government still follow the recommendation of four visits); ii) be assisted by a SBA during delivery; and iii) receive at least one PNC visit within 2 days of delivery from skilled healthcare personnel [23]. Two survey questions regarding ANC services were used to determine if women had received ANC, and if so, the number of times. Delivery care-related questions collected information on who had provided care during delivery and where delivery had occurred. Four additional questions collected information on the timing of the first PNC, as well as providers of PNC. Finally, we combined these ANC, SBA, and PNC variables and created a Continuity of Care (CoC) variable based on previous research in Bangladesh [24]. This variable was categorised as: no CoC (received none of the services), low/moderate level of CoC (received at least one or two of the three services), and high level of CoC (received all three recommended services) [24].

## Adjusted variables

We conducted a three-stage process to identify the variables for adjustment. First, we conducted an extensive literature search for relevant studies in LMICs and Bangladesh [5,13–21,25,26]. Based on this literature review, we generated a list of potential adjustment variables, which were subsequently verified for their presence in the survey dataset. The variables that were found to be available in the survey were then assessed for their statistical significance in relation to the outcome variables. Finally, only the variables that exhibited statistical significance in a forward regression model were considered for inclusion in the analysis. Women's factors included women's current age (15–24 years, 25–34 years, 35–49 years), age at birth of most recent child (≤19 years, 20–34 years, ≥ 35 years), education (no education, primary, secondary, higher), working status (no, yes), decision-making autonomy (continuous variable), month of menstruation resumption (continuous variable), and history of pregnancy termination (yes, no). Husband's age at the time of survey (<24 years, 25–34 years, 35–44 years, ≥ 45 years), husband's education (no education, primary, secondary, higher), parity (≤2, > 2), media exposure (not exposed (i.e., reported no access to radio/television/newspapers within a week of the survey), moderately exposed (i.e., reported at least one day but less than 3 days access to radio/television/newspapers within a week of the survey), and highly exposed (if respondents reported regular access to radio/television/newspapers within a week of the survey)) and pregnancy intention (wanted, unwanted) were household level characteristics included in the model. Other factors considered were place of residence

(urban, rural), wealth quintile (poorest, poor, middle, rich and richest) and administrative divisions (Barishal, Chattogram, Dhaka, Khulna, Mymensingh, Rajshahi, Rangpur, Sylhet).

## Statistical analysis

The characteristics of the study population were described using descriptive statistical analysis. The prevalence of modern FP methods per months of the postpartum period were also determined. A time-to-event approach was used to handle postpartum women, irrespective of whether or not they had adopted modern FP methods by the time of the survey. As such, the 'event' (failure) was postpartum modern contraceptive uptake, and the 'duration' (time) was measured up to the first 12 months (follow-up period) following the most recent live birth. Cumulative probabilities of uptake of any modern FP methods during the postpartum period (0–11 months) was also reported. Kaplan-Meier survival curves were created to illustrate the distribution of any modern FP method use during the postpartum period according to women's socio-demographic characteristics. Finally, a modified poison regression approach was used to determine the effect of maternal healthcare services use on modern FP method use in the postpartum period. A separate model was run for every form of healthcare service used. We used the Poisson modified generalised linear regression model with clustered error variance. Given the survey we analysed had a hierarchical structure and the prevalence of maternal healthcare services use was over 10%, this statistical approach was chosen as it produces a more precise result compared to logistic regression analysis [27]. We ran both unadjusted and adjusted models. In the unadjusted model, each form of maternal healthcare services use was considered with the outcome variable without any confounding variables. In the adjusted model, confounding variables were considered. Before executing the models, we conducted a multicollinearity check utilizing Variance Inflation Factors (VIF) (Supplementary Table 1 in **S1 File**). If we detected evidence of multicollinearity, specifically if the VIF exceeded 10, we proceeded to remove the pertinent variables and then re-ran the models. All analyses accounted for the complex survey design and sampling weights. We used STATA windows version 15.1 MP (StataCorp, LP, College Station, TX, USA) for data analysis.

## Results

### Background characteristics

Table 1 shows the background characteristics of the study participants. Among the 4,081 women included in the study, the majority (70.7%) were aged between 15–24 years at the birth of their most recent child, and 41.0% were aged between 25–34 years. Around half of the women reported having secondary education, 62.7% were classified as unemployed, and 71% had parity ≤ 2. Nearly one-third of the women's partners had completed primary or secondary education. Approximately 73% of the women lived in rural areas, and nearly 26% reported Dhaka as their current residing division.

### Maternal healthcare services utilization

Approximately 44% of the women received at least four ANC from medically trained providers during their most recent pregnancy, 52.9% delivered via a SBA, 49.9% had a facility delivery, and 52% received PNC services (Table 1). The prevalence of CoC was 30.9%.

### Uptake of family planning methods during the postpartum period

Table 2 presents the use of any modern PPFP methods. A total of 71.8% of the analysed sample used modern FP methods within 12 months postpartum. More than one-third (40.2%) initiated these methods from the second month of delivery (Fig 2). The majority of women (39.5%) who

**Table 1. Background characteristics of the ever-married women who have given birth in last three years preceding the survey, BDHS 2017–18 (n = 5012).**

| Characteristics | % (95% CI) |
|---|---|
| **Women's age** | |
| 15–24 years | 53.1 (51.5–54.7) |
| 25–34 years | 41.0 (39.5–42.6) |
| 35–49 years | 5.9 (5.2–6.6) |
| **Age at last childbirth** | |
| ≤19 years | 25.1 (23.7–26.5) |
| 20–34 years | 70.7 (69.3–72.2) |
| ≥35 years | 4.2 (3.7–4.8) |
| **Women's educations** | |
| No education | 6.3 (5.5–7.2) |
| Primary | 27.6 (25.8–29.5) |
| Secondary | 49.0 (47.2–50.8) |
| Higher | 17.1 (15.6–18.7) |
| **Women's working status** | |
| No | 62.7 (60.5–64.8) |
| Yes | 37.3 (35.2–39.5) |
| **Parity** | |
| ≤2 | 71.0 (69.4–72.6) |
| >2 | 29.0 (27.4–30.6) |
| **Husband's age[a]** | |
| <24 years | 7.8 (7.0–8.7) |
| 25–34 years | 50.2 (48.6–51.8) |
| 35–44 years | 34.4 (32.9–35.9) |
| ≥45 years | 7.6 (6.7–8.3) |
| **Husband's education[b]** | |
| No education | 13.6 (12.3–15.2) |
| Primary | 33.6 (31.9–35.4) |
| Secondary | 34.0 (32.4–35.7) |
| Higher | 18.5 (17–20) |
| **Wealth quintile** | |
| Poorest | 20.6 (18.6–22.8) |
| Poorer | 20.5 (19.0–22.1) |
| Middle | 19.2 (17.7–20.8) |
| Richer | 20.2 (18.4–22.0) |
| Richest | 19.5 (17.6–21.6) |
| **Religion** | |
| Non-Muslim | 8.1 (6.6–10.1) |
| Muslim | 91.9 (88.9–93.4) |
| **Media exposure** | |
| Low exposed | 34.3 (31.8–36.7) |
| Moderately exposed | 55.2 (52.9–57.4) |
| Highly exposed | 10.6 (9.5–11.7) |
| **Place of residence** | |
| Urban | 26.9 (25.2–28.6) |
| Rural | 73.2 (71.4–74.8) |

*(Continued)*

**Table 1.** (Continued)

| Characteristics | % (95% CI) |
|---|---|
| **Division** | |
| Barishal | 5.7 (5.1–6.3) |
| Chattogram | 21.2 (19.5–23) |
| Dhaka | 25.6 (23.9–27.4) |
| Khulna | 9.2 (8.3–10.1) |
| Mymensingh | 8.5 (7.7–9.5) |
| Rajshahi | 11.6 (10.4–13) |
| Rangpur | 10.6 (9.5–11.7) |
| Sylhet | 7.6 (6.7–8.5) |
| **Ever had terminated pregnancy** | |
| No | 83.6 (82.4–84.7) |
| Yes | 16.4 (15.3–17.6) |
| **Menstrual resumption** | |
| No | 18.1 (17.0–19.2) |
| Yes | 81.9 (80.8–83.0) |
| **Wanted last child** | |
| Later or no more | 20.9 (19.7–22.1) |
| Wanted then | 79.1 (77.8–80.3) |
| **Maternal healthcare service utilization[c]** | |
| **4 ANC at least one with medically trained providers** | |
| No | 56.3 (54.1–58.5) |
| Yes | 43.7 (41.5–45.9) |
| **Delivery by a skilled birth attendance** | |
| No | 47.1 (44.7–49.5) |
| Yes | 52.9 (50.5–55.3) |
| **Delivery in healthcare facility** | |
| No | 50.1 (47.8–52.4) |
| Yes | 49.9 (47.6–52.2) |
| **Caesarean delivery** | |
| No | 66.8 (64.8–68.7) |
| Yes | 33.2 (31.3–35.2) |
| **Post-natal healthcare services** | |
| No | 47.9 (45.6–50.2) |
| Yes | 52.1 (49.8–54.5) |
| **Level of continuum of care (CoC)** | |
| No CoC | 34.4 (32.2–36.7) |
| Low/moderate level of CoC | 34.7 (33.1–36.4) |
| High level of CoC (WHO recommended level) | 30.9 (28.8–33.1) |

Note: Calculated for column percentages.

[a]missing, n = 146;

[b]missing, n = 141;

[c]was classified according to WHO definition and was measured for last 3 years prior to the survey, sample n = 5012.

[d]Includes the sample who didn't seek ANC or PNC.

**Table 2.** Prevalence of postpartum family planning use by months after childbirth among the ever-married women who have given birth in last three years preceding the survey, BDHS 2017–18 (n = 4081).

| Family planning methods | Months after childbirth, % | | | | | | | | | | | | Total, %[b] |
|---|---|---|---|---|---|---|---|---|---|---|---|---|---|
| | 0th | 1st | 2nd | 3rd | 4th | 5th | 6th | 7th | 8th | 9th | 10th | 11th | |
| **Use of FP methods[a]** | | | | | | | | | | | | | |
| No methods | 100.0 | 0.0 | 0.0 | 0.0 | 0.0 | 0.0 | 0.0 | 0.0 | 0.0 | 0.0 | 0.0 | 0.0 | 20.4 |
| Traditional methods | 0.0 | 1.9 | 34.9 | 22.6 | 14.0 | 3.32 | 4.32 | 8.0 | 2.5 | 3.8 | 3.9 | 0.7 | 7.7 |
| Modern methods | 0.0 | 5.9 | 34.4 | 20.0 | 13.7 | 5.5 | 4.8 | 6.2 | 2.8 | 2.7 | 2.3 | 1.7 | 71.8 |
| **Type of traditional methods[b]** | | | | | | | | | | | | | |
| Safe period | 0.0 | 1.3 | 33.9 | 19.3 | 13.6 | 2.6 | 5.8 | 10.0 | 2.6 | 4.7 | 5.5 | 0.6 | 4.4 |
| Withdrawal | 0.0 | 2.7 | 36.3 | 27.1 | 14.5 | 4.3 | 2.4 | 5.2 | 2.4 | 2.7 | 1.7 | 0.8 | 3.3 |
| **Type of modern methods[b]** | | | | | | | | | | | | | |
| Pill | 0.0 | 30.0 | 1.6 | 36.3 | 22.0 | 13.1 | 4.7 | 5.9 | 3.1 | 3.2 | 2.8 | 2.2 | 39.5 |
| IUD | 0.0 | 48.1 | 24.2 | 9.6 | 5.2 | 0.0 | 6.3 | 3.9 | 2.8 | 0.0 | 0.0 | 0.0 | 0.5 |
| Injectables | 0.0 | 2.0 | 38.6 | 15.4 | 13.2 | 7.1 | 5.4 | 8.0 | 3.2 | 3.0 | 3.0 | 1.1 | 13.1 |
| Condom | 0.0 | 0.7 | 32.4 | 23.0 | 19.7 | 7.3 | 4.5 | 5.8 | 2.1 | 2.3 | 1.2 | 1.0 | 13.2 |
| Female sterilization | 0.0 | 91.2 | 7.3 | 0.0 | 1.0 | 0.0 | 0.0 | 0.5 | 0.0 | 0.0 | 0.0 | 0.0 | 2.8 |
| Male sterilization | 0.0 | 43.8 | 6.8 | 15.5 | 15.5 | 18.3 | 0.0 | 0.0 | 0.0 | 0.0 | 0.0 | 0.0 | 0.2 |
| Implant | 0.0 | 11.2 | 31.2 | 22.3 | 9.1 | 4.4 | 7.3 | 10.8 | 2.3 | 0.0 | 0.0 | 1.3 | 2.4 |
| Other | 0.0 | 33.3 | 0.0 | 33.3 | 0.0 | 33.4 | 0.0 | 0.0 | 0.0 | 0.0 | 0.0 | 0.0 | 0.1 |

Note:

[a]Calculated for cumulative row percentages;

[b]is calculated for column percentage. Other includes all other reported modern contraception (ECP, and others).

reported the use of modern FP methods in the postpartum period used birth control pills, and this usage increased significantly over the observation period. Additionally, 13.1% of women used injectables (Depo-Provera) and 13.2% reported condom use. Female sterilization was reported by 2.8% of women and over 91% was performed in the first month postpartum.

## Cumulative survival probability of using modern family planning methods in the postpartum period

Table 3 and Supplementary Figure 1 in S1 File show the cumulative survival probability of using modern FP methods (for the women who indicated the use of FP methods) in the postpartum period. The probability of using any modern FP methods in the first month postpartum was 6.1%. This significantly increased to 36.9% after the second month. A similar trend was observed in the sixth month, where the percentage of any modern FP method use increased to 80.6%. By the end of the designated postpartum period, the probability of using any modern FP method had risen to nearly 100%. There was no significant variation in modern FP methods used in the postpartum period across women's place of residence (Fig 3) and wealth index (Fig 4). However, we found a variation in modern FP methods used in the postpartum period according to women's level of education, with women with higher education levels more likely to use modern FP methods in the postpartum period than those with no education (Fig 5).

## Association between the maternal healthcare services utilization and uptake of any modern postpartum period family planning methods

Table 4 presents both unadjusted and adjusted associations between maternal healthcare services use and women's use of any modern FP methods in the postpartum period. Full model

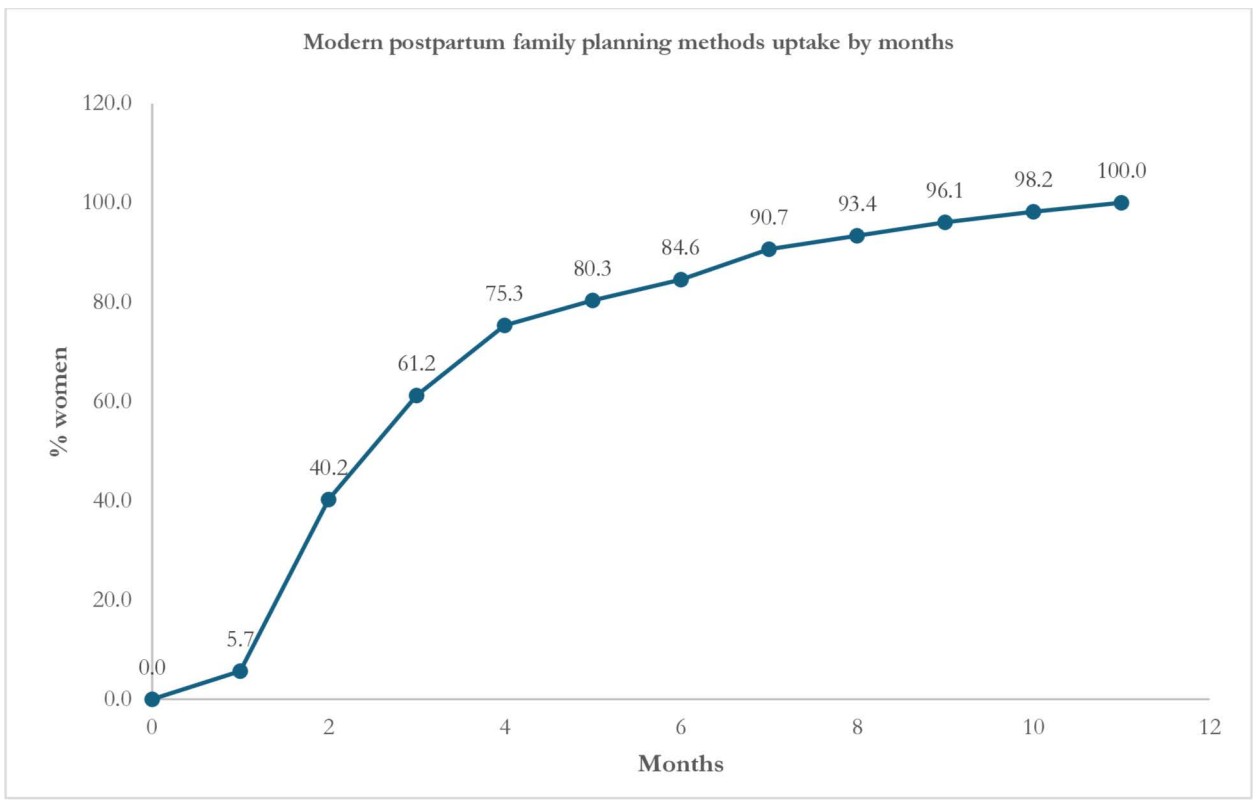

**Fig 2. Cumulative percentage distribution of women who started to use modern postpartum family planning methods, distributed over the months after childbirth (0–11 months).**

**Table 3. Cumulative survivor function to estimate the probability of postpartum modern family planning uptake among women who had given birth in three years preceding the survey, BDHS, 2017–18 (n = 4081).**

| Months | Total n | Failure | Lost | Not using, S(t) | Using, 1-S(t) | Error | 95% CI |
|---|---|---|---|---|---|---|---|
| 0st | 4081 | 46 | 748 | 0.9887 | 0.0113 | 0.0017 | 0.985– 0.9915 |
| 1st | 3287 | 167 | 6 | 0.9385 | 0.0615 | 0.0041 | 0.9299– 0.946 |
| 2nd | 3114 | 1022 | 108 | 0.6305 | 0.3695 | 0.0084 | 0.6138–0.6466 |
| 3th | 1984 | 620 | 77 | 0.4335 | 0.5665 | 0.0087 | 0.4163–0.4505 |
| 4th | 1287 | 418 | 49 | 0.2927 | 0.7073 | 0.0082 | 0.2768–0.3088 |
| 5th | 820 | 148 | 15 | 0.2399 | 0.7601 | 0.0078 | 0.2248–0.2552 |
| 6th | 657 | 125 | 17 | 0.1942 | 0.8058 | 0.0073 | 0.1802–0.2087 |
| 7th | 515 | 181 | 25 | 0.126 | 0.874 | 0.0062 | 0.114– 0.1385 |
| 8th | 309 | 79 | 9 | 0.0938 | 0.9062 | 0.0056 | 0.0832–0.1051 |
| 9th | 221 | 81 | 12 | 0.0594 | 0.9406 | 0.0047 | 0.0507– 0.069 |
| 10th | 128 | 62 | 10 | 0.0306 | 0.9694 | 0.0036 | 0.0242–0.0382 |
| 11th | 56 | 54 | 2 | 0.0011 | 0.9989 | 0.0008 | 0.0002–0.0038 |

Note: S(t): survivor probability of not using contraceptives; 1-S(t): probability of using contraceptives.

results are presented in Supplementary Tables 2–7 in **S1 File**. Women who received at least four ANC visits from skilled providers were found to be 1.04 times more likely (95% CI: 1.00–1.07) to use any modern FP methods in the postpartum period compared to those who did

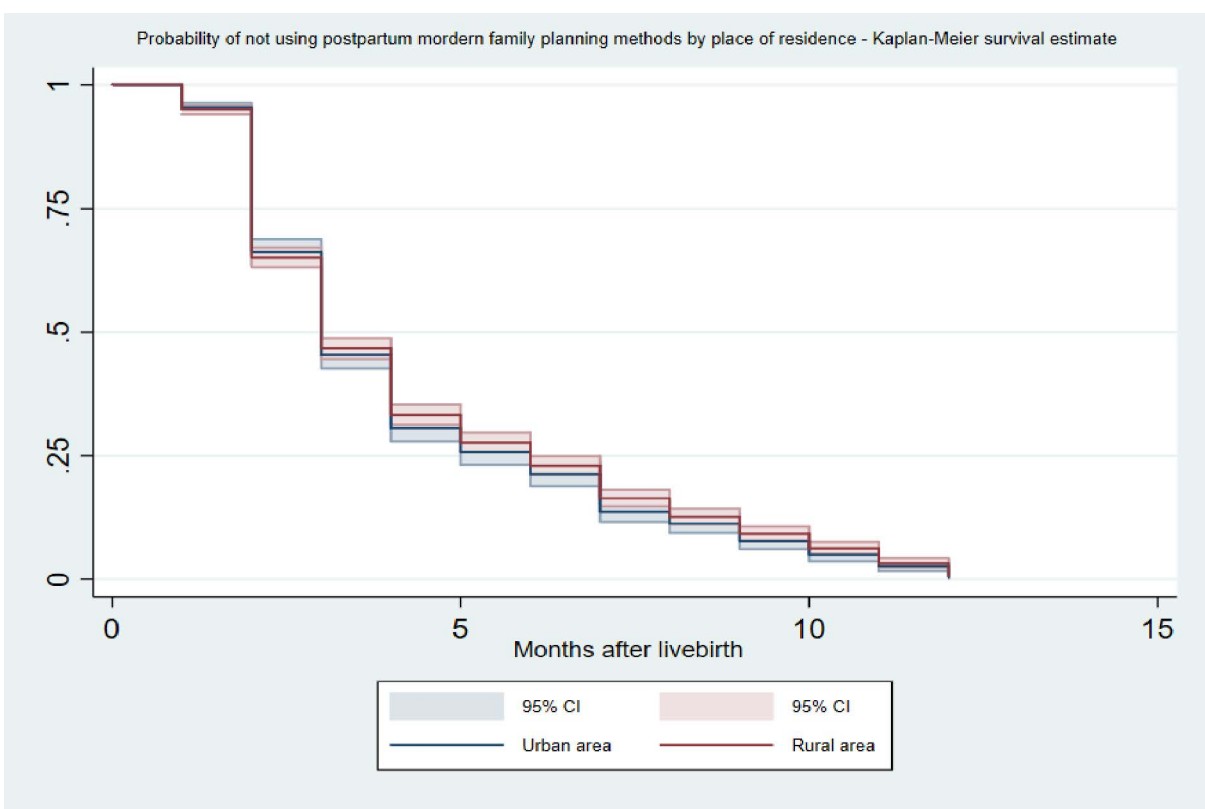

**Fig 3. Probability of not using postpatum modern family planning methods over the months after childbirth (1–12 months) by women's place of residence using Kaplan Meier survival estimates.**

not receive at least four ANC visits. Similarly, the likelihood of using any modern FP methods in the postpartum period was found to be significantly higher for women who delivered with a SBA (aPR: 1.07, 95% CI: 1.03–1.11) and/or at a DHC (aPR: 1.06, 95% CI: 1.02–1.10). Caesarean delivery was also found to be associated with 1.08 times (95% CI: 1.03–1.12) higher likelihood of using any modern FP method in the postpartum period. A significantly higher likelihood of using any modern FP method was found among women who reported the use of PNC (aPR: 1.06, 95% CI: 1.03–1.10) compared to their counterparts who did not report the use of this service. We also found a 1.09 times (95% CI: 1.05–1.13) higher likelihood of using any FP method in the postpartum period among women who reported a higher level of CoC compared to the women who reported lower level of care.

## Discussion

The aim of this study was to explore the association between maternal healthcare service utilization and the adoption of modern FP methods during the postpartum period. Our results indicate that the prevalence of PPFP uptake was 73%. Pill and condom were reported as dominant methods in the postpartum period. The use of maternal healthcare services as well as continuity of using maternal healthcare services, were found to be associated with only a 3%–7% increase in uptake of any modern FP method in the postpartum period. These findings suggest that Bangladesh's national target of counselling on modern FP methods during maternal healthcare services is largely inactive at the field level, and that maternal healthcare service utilization only modestly affects modern FP method adoption during the postpartum

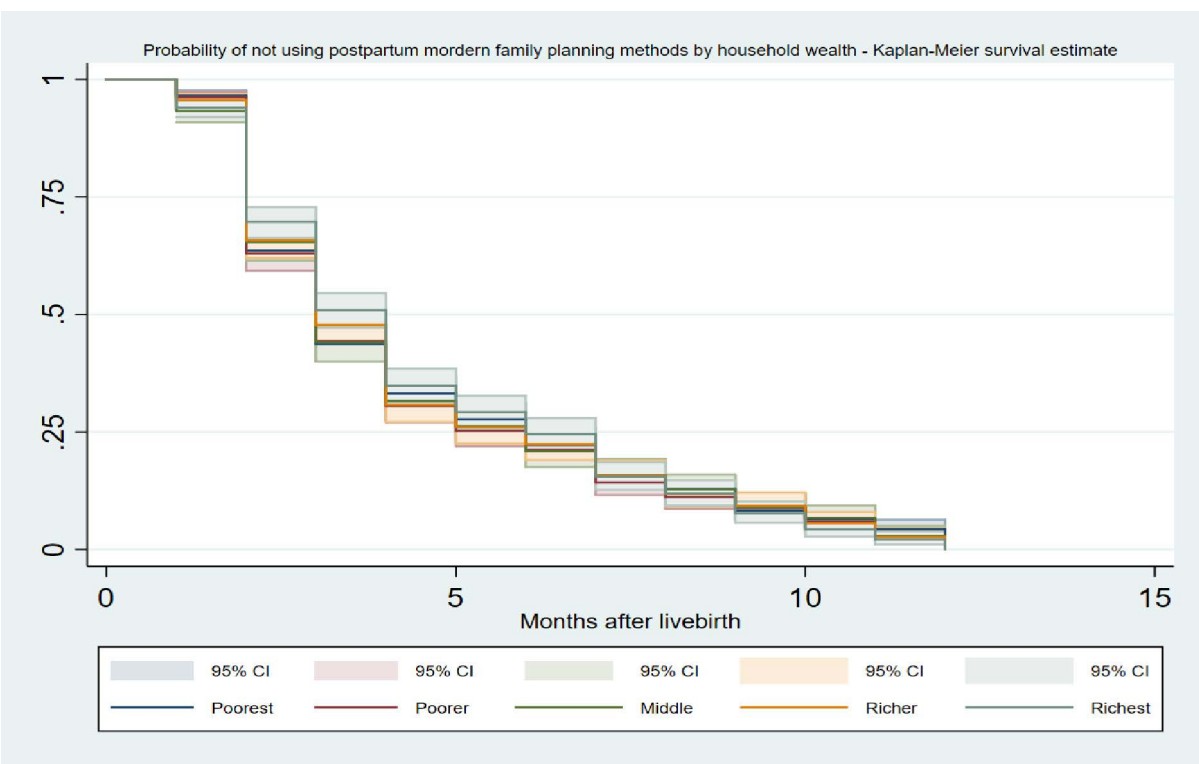

**Fig 4. Probability of not using postpatum modern family planning methods over the months after childbirth (1–12 months) by household wealth using Kaplan Meier survival estimates.**

period. As a result, the government's objectives of reducing unintended pregnancies and short birth intervals by ensuring PPFP through counselling women regarding PPFP during maternal healthcare service usage may not be achieved.

Our findings suggest that over one-quarter of women in Bangladesh do not start using modern FP methods within 12 months after childbirth. Moreover, among women who do use modern FP methods during the postpartum period, the majority do so within the 5th–6th week post-birth, although the prevalence was only 33%, after becoming fertile again [7,28]. This indicates that over 1 million postpartum women in Bangladesh are at risk of a short interval pregnancy, with a significant portion of these pregnancies unintended [9]. In addition, the pattern of modern FP method use during the postpartum period in Bangladesh is problematic. While the postpartum FP program in Bangladesh primarily focuses on providing postpartum IUDs, implants, and female sterilization, these methods account for less than 5% of the total postpartum modern FP methods used in Bangladesh. On the other hand, pills, injections, and condoms are the dominant methods, covering around 92% of the total modern FP method users in Bangladesh in the postpartum period [6,7]. This reflects the general pattern of modern FP method use in Bangaldesh [29]. However, failure rates of these modern methods under typical use conditions are very high, ranging from 4% for injections, to 13–21% for condoms [30]. This places these women at high risk of unintended pregnancies and short birth intervals. Further, progestogen-only pills are primarily recommended for non-breastfeeding women or after six weeks of delivery for breastfeeding women, while condoms are recommended for all women [31]. This pattern of pill use indicates that 35% women are at further risk of adverse child health outcomes given that 65% exclusively breastfeed in

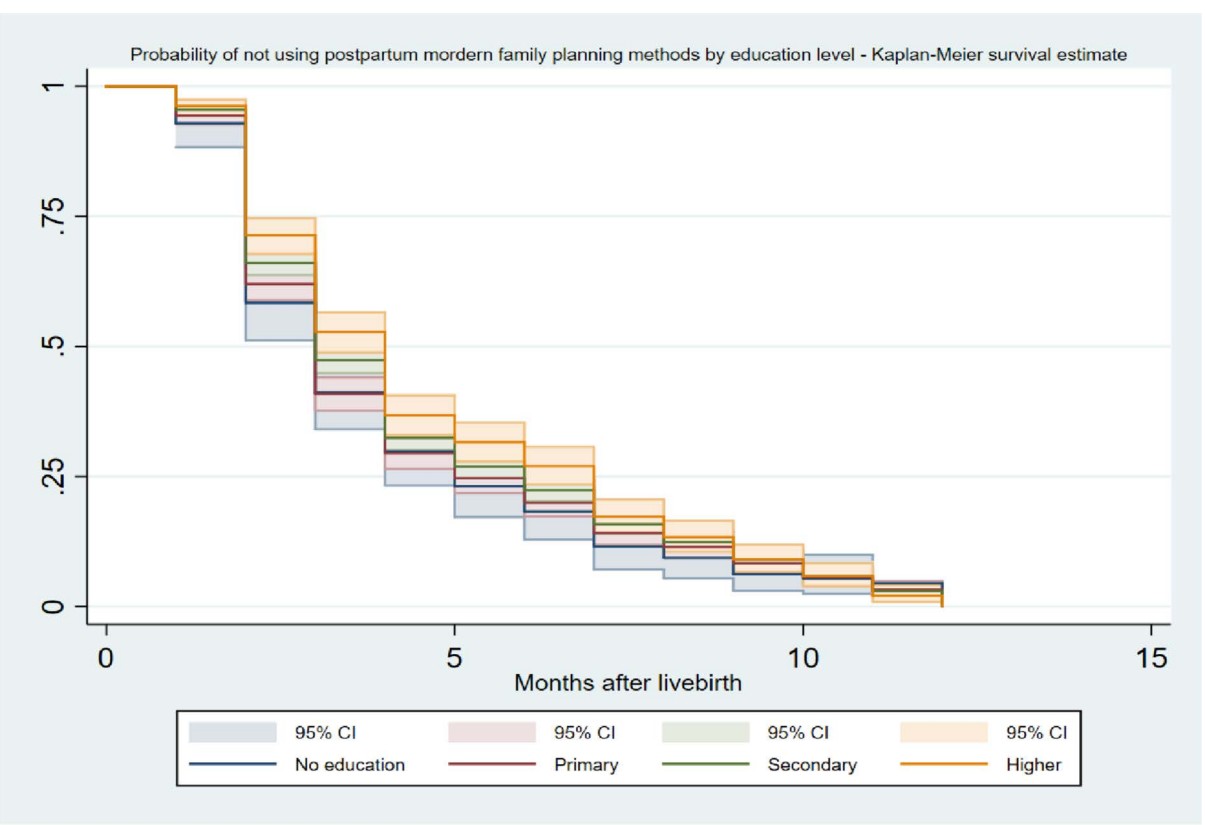

**Fig 5. Probability of not using postpartum modern family planning methods over the months after childbirth (1–12 months) by women's education level using Kaplan Meier survival estimates.**

Bangladesh [7]. Low use of contraception in general and lower use of long-acting modern contraception in particular in Bangladesh may be attributed to insufficient awareness of the effectiveness of these methods to prevent pregnancy during the postpartum period. This may be a result of challenges at both the system and individual level.

Like the WHO's global recommended guidelines, the Government of Bangladesh prioritize women and healthcare personnel contacts during the intrapartum and postpartum periods to provide vital FP counselling [10]. Unfortunately, however, this is not largely reflected at the field level with a lack of proper counselling and poor quality of services identified [32]. Here, we found the use of maternal healthcare service and continuity of care only increased the probability of using modern FP method use in the postpartum period by 3%-7%. While this process has been found effective in several LMICs [13–15,13,33], in Bangladesh, at least three limitations may have attributed to it being less effective. First, the recommendation primarily focuses on postpartum women who have accessed healthcare services; however, over 30% of Bangladeshi postpartum women do not receive any maternal healthcare services during pregnancy [7], similar to women in other LMICs [34,35]. Second, more than 70% of pregnant women in Bangladesh do not access all maternal healthcare services, mainly dropping out after transferring from at least one ANC visit to four or more ANC visits, and from DHC to PNC [24]. A similar trend has been reported in other LMICs [36,37]. In both cases, maternal healthcare services are perceived unnecessary in Bangladesh unless complications arise [24]. Third, while private sectors are becoming increasingly popular in Bangladesh for providing maternal healthcare services, with around 75% of total DHC provided by private facilities,

**Table 4. Association of maternal healthcare service utilization with postpartum modern family planning uptake among women who had given birth in three years preceding the survey using poison modified generalised linear model estimates: BDHS, 2017–18 (n = 4081).**

| Maternal healthcare service utilization | Unadjusted | Adjusted[a] |
|---|---|---|
| | PR (95% CI) | PR (95% CI) |
| **4 ANC at least one with medically trained providers** | | |
| No (ref) | 1.00 | 1.00 |
| Yes | 1.04 (0.99–1.08) | 1.04 (1.00–1.07)* |
| **Delivery by a skilled birth attendance** | | |
| No (ref) | 1.00 | 1.00 |
| Yes | 1.05 (1.01–1.08)* | 1.07 (1.03–1.11)** |
| **Delivery in healthcare facility** | | |
| No (ref) | 1.00 | 1.00 |
| Yes | 1.04 (1.01–1.08)* | 1.06 (1.02–1.10)** |
| **Caesarean delivery** | | |
| No (ref) | 1.00 | 1.00 |
| Yes | 1.06 (1.02–1.09)** | 1.08 (1.03–1.12)*** |
| **Post-natal healthcare services** | | |
| No (ref) | 1.00 | 1.00 |
| Yes | 1.04 (1.01–1.07)* | 1.06 (1.03–1.10)*** |
| **Continuity of care** | | |
| Low (ref) | 1.00 | 1.00 |
| Moderate | 1.01 (0.96–1.06) | 1.02 (0.96–1.07) |
| High | 1.06 (1.01–1.11)* | 1.09 (1.05–1.13)*** |

**Note:**

[a]All the models were run separately for each types of maternal healthcare service utilization and was adjusted for women's menstruation resumption time after childbirth, wanted pregnancy, ever had terminated pregnancy, age at birth, women's education, women's working status, parity (children ever born), husband's age, husband's education, media exposure, wealth quintile and place of residence. Values with superscript asterisks *, **, and ***indicate p < 0.05, p < 0.01, and p < 0.001, respectively. (ref): Reference category, PR: prevalence ratio, CI: confidence interval. Full model results are presented in Supplementary Tables 1–6 in **S1 File**.

the government's policy does not address how they will ensure modern FP method use in the postpartum period among mothers accessing services from any private facilities [10]. Consequently, although maternal healthcare service use is increasing, this does not largely influence modern FP method use in the postpartum period.

These challenges are exacerbated by coordination issues between the two wings of the Ministry of Health and Family Welfare in Bangladesh: the Director General of Family Planning (DGFP) and the Director General of Health Services (DGHS) [10]. Although these two wings have strong parallel infrastructure within Bangladesh and have recently started providing integrated services for FP only, their focuses are different [11]. The DGHS mainly operates maternal healthcare services and immunization, while FP services mainly operated by DGFP. Therefore, DGHS service providers often overlook FP issues because they may not see it as their responsibility [25]. Even if DGHS service providers wish to counsel mothers about modern FP methods, they may not do so because FP is usually considered a culturally sensitive issue in Bangladesh that requires a private room to discuss [25]. However, this is not available in almost all maternal healthcare facilities. These challenges are additional to the DGHS's healthcare providers' higher pressure to treat complicated cases because of very low doctor-population ratio and the lack of monitoring from the governmental level. On the

other hand, DGFP service providers mainly focus on providing FP counselling and distributing contraceptive methods to sexually active, non-pregnant married women [7,25]. They usually do not consider pregnant women and women who have just given birth as their target groups because they do not require contraceptive methods at that specific time [10]. Together, these challenges contribute to the failure of providing appropriate FP counselling to pregnant and postpartum women, despite this being a key target in the national strategy in Bangladesh [26]. However, the few instances of counselling that are provided during maternal healthcare service use have shown modest effects on the uptake of FP methods in the postpartum period, as reported in this study as well as other studies conducted in LMICs [13,31,33].

At an individual level, limited knowledge of FP methods, as well as the stigma and cultural norms surrounding it play important roles in influencing the uptake of FP methods among postpartum women [26]. For instance, a significant proportion of postpartum women in Bangladesh lack knowledge regarding the importance of using FP methods during the postpartum period, and there is a belief in the community that women do not need FP methods following delivery, particularly while breastfeeding [7,26]. These difficulties are compounded by limited access to information and services related to family FP methods, which is increasing rather than declining. For instance, exposure to FP messages through home visits by family planning workers decreased from 40% in 1994 to 13% in 2017/18, and exposure through mass media decreased from 50% in 1994 to 24% in 2017/18 [7,10]. The reasons for this could be the ongoing lack of FP workers at the community level, inadequate monitoring, and extra burden to provide services to more people, as the current number of posts for family planning workers was developed in early 1978 when the total population of Bangladesh was less than half of what it is now [11]. Moreover, a significant number of the currently approved posts are also vacant [25]. Additionally, women are increasingly outside the home for work may reduce their availability at home when visited by family planning workers [26]. Together, these factors may contribute to the lower uptake of postpartum FP methods.

The findings of this study suggest important policy implications for increasing the uptake of PPFP. It is evident that the current approach to increasing PPFP through access to maternal healthcare services is not as effective as desired; their role is relatively minor but still significant. Therefore, there is an urgent need to address the challenges within maternal healthcare sector to enhance counselling on PPFP. This includes increasing the coverage of maternal healthcare services and the number of healthcare personnel dedicated to provide maternal healthcare. Furthermore, PPFP counselling should be integrated as a vital component of the services provided by private healthcare facilities in Bangladesh. Challenges associated with providing and ensuring maternal healthcare services and FP in government healthcare facilities also need to be addressed. This involves ensuring proper integration between the DGHS and the DGFP and addressing structural issues, such as designating specific areas for PPFP services.

This study has several strengths and a few limitations. First, we analysed a large sample extracted from a nationally representative survey. The results are adjusted with a wide range of confounding variables at the individual, household, and community level through advanced statistical modelling. The WHO recommendations were followed to generate modern FP methods variables, and maternal healthcare service use variables. Together, these allowed us to generate more precise findings. However, analysis of cross-sectional data is the primary limitation of this study; therefore, the findings reported in this study are cross-sectional only, not causal. Additionally, both maternal healthcare services and contraception uptake data, as reported in the calendar methods, were collected retrospectively. Consequently, there is a potential for recall bias, although any such errors are likely to be random in nature. Moreover, rather than the factors we considered in the model, community-level norms and traditions

and availability and readiness of healthcare facilities play an important role in ensuring FP methods uptake during the postpartum period. However, we could not do it because of the lack of relevant data. Furthermore, the global literature has demonstrated the effectiveness of four postnatal care visits in increasing the adoption of modern FP methods. Unfortunately, we were unable to consider this factor due to a lack of available data. Another limitation is our inability to consider unmarried women in the analysis as the survey data we analysed did not collect this information considering Bangaldeshi culture is prohibitive of sex outside of marriage and evidence of this is mostly unreported. Despite these limitations which needs further exploration, as far as we know, this is the first study of its kind, which should be used for making country-level policies and programs to ensure FP methods uptake at the postpartum period. Future research should be conducted by implementing a prospective study design along with a broad range of contributing factors.

## Conclusion

Modern FP methods in the postpartum period were reported by 73% of women. However, over 60% started using modern FP methods six months post-birth. Along with these challenges, the pattern of modern FP methods reported in this study was found to be problematic, with very few (<5%) using modern FP methods that are recommended for postpartum women. Added to this, maternal healthcare service use had only modest effects on up taking modern FP methods. These findings indicate that the 2015 National Action Plan for PPFP in Bangladesh is not active enough at the field level in Bangladesh. This might contribute to a higher occurrence of short interval births and unintended pregnancies and related adverse outcomes, including maternal and child mortality. This would challange Bangladesh's ability to achieve the SDGs targets related to improving maternal and child health. Enhancing maternal healthcare services, integrating PPFP as a component of services offered by private healthcare facilities, and addressing structural challenges are recommended to improve the uptake PPFP. This however requires a greater focus on the healthcare service delivery system.

## Supporting information

**S1 File. Supplementary file.**
(DOCX)

## Acknowledgments

We are thankful to MEASURE DHS for the data support. Also we are grateful to icddr,b which is grateful to the Governments of Bangladesh, Canada, Sweden and the UK for providing core/unrestricted support for its operations and research, where the data for this study was analysed. The authors also acknowledge the support of Maternal and Child Health Division of icddr,b, Health System and Population Studies Division of icddr,b and Department of Population Science of Jatiya Kabi Kazi Nazrul Islam University, where this study was designed and conducted.

## Author contributions

**Conceptualization:** Md. Nuruzzaman Khan, Md. Moinuddin Haider.

**Data curation:** Md. Nuruzzaman Khan, Md. Mostaured Ali Khan, Md Arif Billah, Md. Moinuddin Haider.

**Formal analysis:** Md. Nuruzzaman Khan, Md. Mostaured Ali Khan, Md Arif Billah.

**Funding acquisition:** Md Arif Billah.

**Investigation:** Md. Mostaured Ali Khan, Md. Moinuddin Haider, Bidhan Krishna Sarker.

**Methodology:** Md. Nuruzzaman Khan, Shimlin Jahan Khanam.

**Supervision:** Bidhan Krishna Sarker, Melissa L. Harris.

**Writing – original draft:** Md. Nuruzzaman Khan.

**Writing – review & editing:** Md. Nuruzzaman Khan, Md. Mostaured Ali Khan, Md Arif Billah, Shimlin Jahan Khanam, Melissa L. Harris.

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
