## [Decision Letter · Decision Letter 0]

20 Sep 2023

PONE-D-23-19771Maternal Healthcare Service Utilization and Modern Postpartum Family Planning Access in Bangladesh: Insights from a National Representative SurveyPLOS ONE

Dear Dr. Khan,

Thank you for submitting your manuscript to PLOS ONE. After careful consideration, we feel that it has merit but does not fully meet PLOS ONE’s publication criteria as it currently stands. Therefore, we invite you to submit a revised version of the manuscript that addresses the points raised during the review process.

We look forward to receiving your revised manuscript.

Kind regards,

Temesgen Tilahun

Academic Editor

PLOS ONE

Reviewers' comments:

Reviewer's Responses to Questions

**Comments to the Author**

1. Is the manuscript technically sound, and do the data support the conclusions?

Reviewer #1: Partly

Reviewer #2: Yes

Reviewer #3: Yes

2. Has the statistical analysis been performed appropriately and rigorously? 

Reviewer #1: No

Reviewer #2: Yes

Reviewer #3: Yes

3. Have the authors made all data underlying the findings in their manuscript fully available?

Reviewer #1: Yes

Reviewer #2: Yes

Reviewer #3: Yes

4. Is the manuscript presented in an intelligible fashion and written in standard English?

Reviewer #1: Yes

Reviewer #2: Yes

Reviewer #3: Yes

5. Review Comments to the Author

Reviewer #1: Did the assumptions of the model fit? Effect modification? Multi-collinearity?

Provide residual plots of the assumption.

If the variables were not significant in the unadjusted model, why then adjusted in the multiple model?

In multivariable regression, variable selection requires justification like check of confounding, interaction, and association in bivariate regression along with the support of the literature.

After fitting the model, did author check the model fitness?

Report AIC, BIC.

Post estimation test, VIF require to do to check for the collinearity.

The ROC curve with the AUC should be reported to make sure that the models predicted the outcome.

"As such, the ‘event’ (failure) was

postpartum modern contraceptive uptake, and the ‘duration’ (time) was measured up to the first

12 months (follow-up period) following the most recent live birth"

in BDHS event and time are measured at the same time (single interview), so, the duration is not possible to measure as the temporality is not exist (it is ecological). so, the attempt of using KM may not have the luxury of validity. The author needs to address this and provide a transparent explanation.

Reviewer #2: Title is original and specific but not time bound. Results discussion and conclusion are baseon data generated and well written

Tables need to be submitted separately

Overall it is good research in short of minor problems which could be easily edited

Reviewer #3: Title: Your title clear and self-explanatory

Abstract

-The abstract is well written

-Avoid abbreviation and acronyms from abstract

Introduction

-Well written, except some grammar error

Methods:

The methodology part is well stated with some concerns

-the way your screened the study participants were clear and self-explanatory

- You classified your outcome variable as used modern family planning during post-partum period, even the post-partum period you defined was highly extended. The contraceptive methods you classified as modern family planning were short acting (POP, COC, Depo-Provera) Long acting (Implants (Norplant) and sterilization (female and male sterilization).

How you see the Norplant methods? Currently, WHO do not recommend norplant contraceptive in steady, there are safe and effective as Norplant are available like Jeddele, Sinoplant and implanon). Please clearly state if there are no other implants were not available in Bangladesh during this study period, unless your source of data is under question.

What about Lactational Amenorrhea Method(LAM)?

Are you considered LAM method?

Exposure variables

Utilization of maternal healthcare services during pregnancy, child birth and post-partum period are obviously known as it increases post-partum utilization of family planning. No need to prove by research. Better if you saw its effectiveness.

It is too vague to see all maternal health service as one exposure variable.

When we see ANC has four visits according to your study, currently, WHO recommended eight contacts. In your study, number of visit not considered at what visit post family planning is focused?

The very important exposure component of maternal health service is post-partum care specially fourth visit of post-partum care. Why you considered post-partum visit within two days mean first visit? Other components of maternal health service have on impact on post-partum contraceptive use. Why you categorized mode of delivery as a component of maternal health service (Delivery at health facility and Caesarian section)?

Adjusted variable- How you classify women’s working status as Yes/ No response?

Analysis methods and models you used is interested – Poison regression model is appropriate for this study.

Result

Result part is well written including tables and figures.

Discussion

-lacks comparing with the existing data/ literatures

-lacks justifications

-Needs clearly stating the implication of your finding

Conclusion

Please conclude your finding with the existing context

# Please, revise all the documents regarding grammar and spelling error

6. PLOS authors have the option to publish the peer review history of their article (what does this mean? ). If published, this will include your full peer review and any attached files.

**Do you want your identity to be public for this peer review?** For information about this choice, including consent withdrawal, please see our Privacy Policy .

Reviewer #1: **Yes: ** Masuda Akter

Reviewer #2: **Yes: ** Dr. Lammii Gonfaa Dinagde(Assistant professor of Obstetrics and Gynecology)Wallagaa University Nekemte)

Reviewer #3: No

---

## [Author Response · Author response to Decision Letter 0]

31 Oct 2023

A MS word file addressing all reviewers' comments is uploaded to the system.

---

## [Decision Letter · Decision Letter 1]

12 Feb 2024

PONE-D-23-19771R1Maternal Healthcare Service Utilization and Modern Postpartum Family Planning Access in Bangladesh: Insights from a National Representative SurveyPLOS ONE

Dear Dr. Khan,

Thank you for submitting your manuscript to PLOS ONE. After careful consideration, we feel that it has merit but does not fully meet PLOS ONE’s publication criteria as it currently stands. Therefore, we invite you to submit a revised version of the manuscript that addresses the points raised during the review process.

**ACADEMIC EDITOR: **

Dear authors,

I kindly request that you submit in the revised version of your manuscript that takes into account every criticism raised by the reviewers.

We look forward to receiving your revised manuscript.

Kind regards,

Temesgen Tilahun

Academic Editor

PLOS ONE

Reviewers' comments:

Reviewer's Responses to Questions

**Comments to the Author**

1. If the authors have adequately addressed your comments raised in a previous round of review and you feel that this manuscript is now acceptable for publication, you may indicate that here to bypass the “Comments to the Author” section, enter your conflict of interest statement in the “Confidential to Editor” section, and submit your "Accept" recommendation.

Reviewer #4: All comments have been addressed

Reviewer #5: (No Response)

Reviewer #6: (No Response)

2. Is the manuscript technically sound, and do the data support the conclusions?

Reviewer #4: Yes

Reviewer #5: No

Reviewer #6: Yes

3. Has the statistical analysis been performed appropriately and rigorously? 

Reviewer #4: Yes

Reviewer #5: No

Reviewer #6: No

4. Have the authors made all data underlying the findings in their manuscript fully available?

Reviewer #4: Yes

Reviewer #5: No

Reviewer #6: Yes

5. Is the manuscript presented in an intelligible fashion and written in standard English?

Reviewer #4: Yes

Reviewer #5: Yes

Reviewer #6: Yes

6. Review Comments to the Author

Reviewer #4: (No Response)

Reviewer #5: Major comment

• Your title should be modified in line with your objective, dependent variable and study protocol

• Your back ground part; result and conclusion part mainly talks to family planning only where other maternal health care service is because it is huge.

• Start your background part with statement of maternal health care service introduction then burden and its prevalence global to local context, strategies to overcome problems…. Flow

• In your study participant inclusion criteria what about unmarried but uses FP and maternal service

• What is your study objective?

• What is your dependent variable?

• What was your study design

• What was your study period?

• Is your study homogenous?

• Please rewrite your result part to show clearly what socio-demographic part, obstetric and gynecological history….

• Your discussion, conclusion and recommendation should be in line with your finding only

• Ethical consideration needed to use such like data

• Put brief description of study area

Reviewer #6: Manuscript Number: PONE-D-23-19771R1

Surely, I would be pleased to assess this work that is being considered for publication in Plosone.

To editor �I think this paper is also presented in “the lancet library”
https://papers.ssrn.com/sol3/papers.cfm?abstract_id=4463501 and check for grammatical issue (I could not access word doc.)

Comments to Author(s)

General comments

1. Dear author, I tried to find what by mean “modern family planning” through your document� please operationalize it

2. There is no line number for your document to comment it by line

Abstract

1. Your background does not give any clues towards effects of using modern family planning or lack of it�modify it

2. Your way of reporting “result” is good�add contributing/associated factors here with its appropriate statistics

3. Put your keywords in alphabetical order

Methods

1. Dear Author, you believe that information gathered six years ago can be used to draw conclusions that policymakers could use it

2. What is your criteria to categorize media exposure to “no, moderate & highly exposed” (has no sense)???

3. Page 10 , line 1& 2�you should mention statistical tool/ software you have used rather than what you did (this should be result part)

Discussion

1. Result & discussion well written� in discussion you did not discuss about contributing factors ----why????

2. Conclusion � so what should will happen or for whom/what is your recommendation

Thank you!

7. PLOS authors have the option to publish the peer review history of their article (what does this mean? ). If published, this will include your full peer review and any attached files.

**Do you want your identity to be public for this peer review?** For information about this choice, including consent withdrawal, please see our Privacy Policy .

Reviewer #4: **Yes: ** Ahmed Afifi

Reviewer #5: No

Reviewer #6: No

---

## [Author Response · Author response to Decision Letter 1]

26 Feb 2024

We have added a MS word file where we provided a point by point response to each of the reviewers' comments.

---

## [Decision Letter · Decision Letter 2]

8 Jul 2024

PONE-D-23-19771R2

Effects of Maternal Healthcare Service Utilization on Modern Postpartum Family Planning Access in Bangladesh: Insights from a National Representative Survey

PLOS ONE

Dear Dr. Khan,

Thank you for submitting your manuscript to PLOS ONE. After careful consideration, we have decided that your manuscript does not meet our criteria for publication and must therefore be rejected.

I am sorry that we cannot be more positive on this occasion, but hope that you appreciate the reasons for this decision.

Kind regards,

Abiodun Adanikin, Ph.D

Academic Editor

PLOS ONE

Additional Editor Comments:

The objective stated in the introduction does not seem to align with the selected data source. Utilizing the 2017-18 Bangladesh Demographic Health Survey (BDHS) will not provide specific insights into the success of Bangladesh's National Action Plan for PPFP, which began in 2015, as suggested in the introduction. Some women captured in the 2017-18 DHS may have given birth shortly before or around the time the Action Plan was launched in 2015. As a result, they may not be the ideal population to assess the plan's success. The research rationale and message should be realigned. Alternatively, a more recent DHS could be used to indirectly provide insight into the plan's success, which began in 2015.

Reviewers' comments:

Reviewer's Responses to Questions

**Comments to the Author**

1. If the authors have adequately addressed your comments raised in a previous round of review and you feel that this manuscript is now acceptable for publication, you may indicate that here to bypass the “Comments to the Author” section, enter your conflict of interest statement in the “Confidential to Editor” section, and submit your "Accept" recommendation.

Reviewer #7: All comments have been addressed

2. Is the manuscript technically sound, and do the data support the conclusions?

Reviewer #7: Yes

3. Has the statistical analysis been performed appropriately and rigorously? 

Reviewer #7: Yes

4. Have the authors made all data underlying the findings in their manuscript fully available?

Reviewer #7: Yes

5. Is the manuscript presented in an intelligible fashion and written in standard English?

Reviewer #7: Yes

6. Review Comments to the Author

Reviewer #7: Introduction: Briefly mention successful interventions for PPFP integration within maternal healthcare services in other LMICs (line 76).

You could condense some information in paragraphs 48-58 to avoid redundancy.

Methods: Consider adding a brief sentence at the beginning of the Methods section to introduce it.

In the Outcome Variable section (paragraph 142), you can mention the specific question used in the survey to identify types of FP methods used.

Discussion: Briefly mention the study objectives at the beginning of the discussion section to refresh the reader's memory.

Consider adding a transition sentence between the first paragraph (findings) and the second paragraph (implications).

When discussing limitations, elaborate more on how they might affect the generalizability of the findings.

7. PLOS authors have the option to publish the peer review history of their article (what does this mean? ). If published, this will include your full peer review and any attached files.

**Do you want your identity to be public for this peer review?** For information about this choice, including consent withdrawal, please see our Privacy Policy .

Reviewer #7: **Yes: ** Mariyam Sarfraz

- - - - -

---

## [Author Response · Author response to Decision Letter 2]

26 Jul 2024

I have uploaded relevant files- please see the attached.

---

## [Decision Letter · Decision Letter 3]

15 Jan 2025

Effects of Maternal Healthcare Service Utilization on Modern Postpartum Family Planning Access in Bangladesh: Insights from a National Representative Survey

PONE-D-23-19771R3

Dear Dr. Khan,

We’re pleased to inform you that your manuscript has been judged scientifically suitable for publication and will be formally accepted for publication once it meets all outstanding technical requirements.

Kind regards,

Akaninyene Eseme Bernard Ubom, MBBS, MSc (Reproductive Physiology), FWACS (ObGyn)

Academic Editor

PLOS ONE

Additional Editor Comments (optional):

Reviewers' comments:

Reviewer's Responses to Questions

**Comments to the Author**

1. If the authors have adequately addressed your comments raised in a previous round of review and you feel that this manuscript is now acceptable for publication, you may indicate that here to bypass the “Comments to the Author” section, enter your conflict of interest statement in the “Confidential to Editor” section, and submit your "Accept" recommendation.

Reviewer #7: All comments have been addressed

2. Is the manuscript technically sound, and do the data support the conclusions?

Reviewer #7: Yes

3. Has the statistical analysis been performed appropriately and rigorously? 

Reviewer #7: Yes

4. Have the authors made all data underlying the findings in their manuscript fully available?

Reviewer #7: Yes

5. Is the manuscript presented in an intelligible fashion and written in standard English?

Reviewer #7: Yes

6. Review Comments to the Author

Reviewer #7: All comments have been addressed. The manuscript is reformatted appropriately, with the appropriate tables and references.

7. PLOS authors have the option to publish the peer review history of their article (what does this mean? ). If published, this will include your full peer review and any attached files.

**Do you want your identity to be public for this peer review?** For information about this choice, including consent withdrawal, please see our Privacy Policy .

Reviewer #7: **Yes: ** Mariyam Sarfraz

---

## [Editor Report · Acceptance letter]

PONE-D-23-19771R3

PLOS ONE

Dear Dr. Khan,

I'm pleased to inform you that your manuscript has been deemed suitable for publication in PLOS ONE. Congratulations! Your manuscript is now being handed over to our production team.

Kind regards,

on behalf of

Dr. PLOS Manuscript Reassignment

Staff Editor

PLOS ONE